# Capturing human categorization of natural images by combining deep networks and cognitive models

Ruairidh M. Battleday [1,3✉], Joshua C. Peterson [1,3✉] & Thomas L. Griffiths[1,2]

Human categorization is one of the most important and successful targets of cognitive modeling, with decades of model development and assessment using simple, low-dimensional artificial stimuli. However, it remains unclear how these findings relate to categorization in more natural settings, involving complex, high-dimensional stimuli. Here, we take a step towards addressing this question by modeling human categorization over a large behavioral dataset, comprising more than 500,000 judgments over 10,000 natural images from ten object categories. We apply a range of machine learning methods to generate candidate representations for these images, and show that combining rich image representations with flexible cognitive models captures human decisions best. We also find that in the high-dimensional representational spaces these methods generate, simple prototype models can perform comparably to the more complex memory-based exemplar models dominant in laboratory settings.

[1] Department of Computer Science, Princeton University, 35 Olden Street, Princeton, New Jersey 08540, USA. [2] Department of Psychology, Princeton University, South Drive, Princeton, New Jersey 08540, USA. [3]These authors contributed equally: Ruairidh M. Battleday, Joshua C. Peterson. ✉email: ruairidh.battleday@gmail.com; peterson.c.joshua@gmail.com

The problem of categorization—how an intelligent agent should group stimuli into discrete concepts—is an intriguing and valuable target for psychological research: it extends many influential themes in Western classical thought[1], has clear interpretations at multiple levels of analysis[2], and is likely fundamental to understanding human minds and advancing artificial ones[3]. Psychological theories of categorization began with the notion that people learn rules or definitions for categories[4,5], but the work of Rosch et al.[6,7] in the 1970s challenged this view by showing that natural categories lacked defining features and were better characterized in terms of a form of family resemblance structure. This led researchers to explore theories in which categories are represented by an abstract prototype, where evaluating the similarity to this prototype could account for behavioral phenomena such as differences in stimulus typicality under a category[8], preference for the (unseen) prototype in recognition tasks[9], and the apparent graded nature of category structure[10]. In the same decade, a competing account was offered in which the same behavioral phenomena could be captured by calculations performed solely on the known members—or, exemplars—of a category, without reference to an abstract prototype[11].

This debate has been distinctive for the role that mathematical modeling and high-precision behavioral experiments have played in working towards resolving it. Prototype models can be formalized by assuming that category membership is calculated using a decision rule based on the similarity to the prototypes of candidate categories[12]. Exemplar models, by contrast, base this categorization decision on the summed similarity to all known members of a category[11,13]. Formalizing these models makes it possible to design experiments that distinguish between them, often by constructing sets of purposefully novel stimuli that result in different predictions for category membership under different models. Laboratory findings in this tradition have largely favored exemplar models, which can accommodate arbitrarily complex boundaries between categories[14,15], demonstrating that people can learn such boundaries when given sufficient training[13,16–18]. Beyond straightforward assessments of categorization accuracy, exemplar models have a number of other advantages. As exemplars are stored, they provide a common substrate for both categorization and recognition memory[19], can be leveraged in process-level accounts of categorization[20], and could theoretically be used for later learning or generating abstractions on-the-fly[21].

Although this work has been insightful and theoretically productive, we know little about how it relates to the complex visual world it was meant to describe: the focus on designing experiments to distinguish between models means that it derives almost exclusively from studies using highly controlled and simplified perceptual stimuli, represented mathematically by low-dimensional hand-coded descriptions of obvious features or low-dimensional multidimensional-scaling (MDS) solutions based on similarity judgments[11,12,17,22–31] (see Fig. 1). Extending these findings to more realistic settings, and in particular to natural images, remains a central challenge. There are two compelling reasons to take this up. The first concerns ecological validity: human categorization abilities emerge from contact with the natural world and the problems it poses; the category divisions that result may be best understood in this context. The use of naturalistic stimuli is common in the related fields of object and scene recognition[32–36] and face perception[37–41]; however, the modeling frameworks they employ are typically inspired by neuroscience and computer vision, making it difficult to connect their findings back to cognitive psychology. Our contribution here is to give a theoretical and empirical analysis of formal cognitive models of categorization on a large and well-known set of natural images that should allow better statistical and experimental integration with these fields. In this sense, it complements recent work that uses exemplar models to capture behavior over a narrower subset of natural images—geological samples—which can be more easily described in low-dimensional spaces[42,43]. The second motivation is theoretical: the different mathematical forms of prototype and exemplar strategies imply their performance will depend differently on how stimuli are represented. In particular, the effect of increasing the dimensionality of stimulus representations on categorization model performance is largely unexplored and yet likely to be a key factor in adequately representing more complex natural stimuli.

In this work, we take a step towards making this extension by using a range of modern supervised and unsupervised computer vision methods to estimate stimulus structure of more complex and naturalistic images, which can then be used as the basis for cognitive modeling of categorization behavior. These methods, and in particular convolutional neural networks (CNNs), have been applied with great recent success on the related task of natural image classification, under the complementary strategy of learning better feature spaces for complex naturalistic stimuli[44]. Although it is unclear to what extent CNN classification models resemble human categorization or feature learning, two properties make them a promising source of representations for modeling human categorization behavior. The first is that these networks are trained on extremely large datasets of natural images, implying that they generalize broadly and offer a greater chance of approximating the experience an individual might have had with a particular class of natural objects. As seeking such a high degree of statistical approximation is likely the closest we can come to matching an individual's real set of exemplars for naturalistic images, the ever-increasing size of training datasets supports using such models to provide stimulus embeddings. The second is that, given we cannot access human mental representations directly, these CNN representations have been shown to offer surprisingly good proxies for them in predicting visual cortex brain activity[45,46] and in psychological experiments investigating similarity judgments, which are closely related to categorization[47,48].

To evaluate models of human categorization of natural images using these representations, we collect and present a large behavioral dataset of human categorizations, which we call CIFAR-10H. This dataset comprises over 500,000 classifications of 10,000 natural images from ten categories from the test subset of the CIFAR-10[49] benchmark dataset that is widely used in computer vision (see Fig. 2). As these images have been extensively explored by the machine learning community, they come with a wealth of corresponding feature sets that may be used as the representational basis for our psychological models. Furthermore, the size and resolution of the dataset allow us to increase the flexibility of our categorization models so that they can further adapt the underlying high-dimensional representations to better capture and more finely assess graded category membership over stimuli. With this dataset in hand, we are able to extend the assessment of cognitive models of human categorization to large and varied numbers of more naturalistic stimuli. We find that choice of feature representation affects the predictive performance of categorization models profoundly, which cognitive models are of most benefit over their machine learning counterparts for ambiguous images, and that there is little difference in the performance of prototype and exemplar strategies in the types of high-dimensional representational spaces that support natural image categorization best.

## Results

**A naturalistic image dataset.** Our CIFAR-10H behavioral dataset consists of 511,400 human categorization decisions made

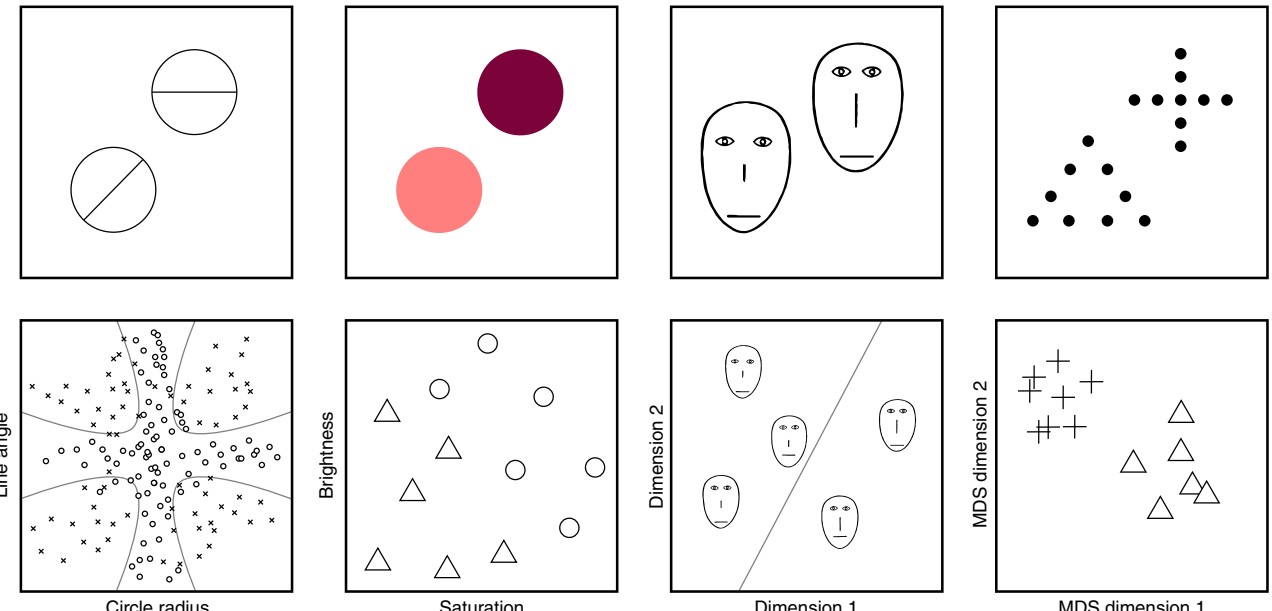

**Fig. 1 Stimuli from previous seminal studies of categorization.** The top row shows representative stimuli, the bottom shows the types of stimulus representations used as input to categorization models. Stimuli and category distributions reproduced to roughly replicate those used in the following studies: left[17], center left[28], center right[12], right[30] (although for the center right panel, stimulus distributions were represented—and linearly separable—in four dimensions[12]).

over 10,000 natural images and collected via Amazon Mechanical Turk. Our image stimuli were taken from the test subset of the CIFAR-10 dataset, which consists of 1000 images for each of the following ten categories: airplane, automobile, bird, cat, deer, dog, frog, horse, ship, and truck. Participants ($N = 2570$) were shown Lanczos-upsampled $160 \times 160$-pixel images and were asked to categorize each image as quickly and accurately as possible (see Fig. 3a). After successful practice, each participant categorized 20 images from each category for the main experiment phase, yielding ~50 judgments per image (see Fig. 3b). In Fig. 2, we use these category judgments to present our images so that their global and local similarity structure is preserved according to our behavioral data. Based on the category judgments for each image, we embed them into a $100 \times 100$ two-dimensional grid such that adjacent images are nearest neighbors according to their category judgments.

We chose the above image set for two reasons. First, we consider the current work to be a conceptual step towards modeling categorization over natural stimuli and so have chosen images that retain some relevance to previous seminal work—for example, their simple and well-delineated nature (see Fig. 1). Notably, however, these stimuli are significantly more varied in nature, both within and between categories, than artificial ones, exhibiting variability that more closely corresponds to the natural world. Although our common linguistic labels at the basic level of Rosch's taxonomy[50] differ from existing laboratory work—which is best placed at the subordinate level—the variability the dataset affords means that there are many instances in which distinguishing between a pair of images—of a bird and plane in the sky, say—involves categorization using a subset of features that parallels this previous work. The pairs of categories for which this is especially true is evident in our confusion matrix (see Fig. 3d). Further motivation for the use of these images rather than more simple objects or object sketches comes from the computer vision and machine learning communities. Although classification algorithms have already been able to correctly classify sketch drawings of natural objects better than humans[51,52], for our CIFAR-10 images the state-of-the-art

classifiers and humans are approximately equally accurate, despite nearly a decade of active investigation. Therefore, our images represent a well-supported point in the tradeoff between successful representational bases and the difficulty of images and category structure, and complement another recently developed natural image dataset that focuses more on scaling the number of categories, as opposed to the number of images and judgments[42,43]. Second, CIFAR-10 is the natural candidate from within contemporary natural image benchmarks in computer vision, with a long and still active history of exploration by the machine learning community. This means that in addition to the range of representations already available, it is likely that the baseline representations and any innovations from the present work will continue to improve the fit to humans. The number of images is small enough to collect enough human judgments to offer a good approximation of the underlying population-level guess distribution for the entire test subset (which machine learning algorithms are not trained on) and the low resolution of the images has advantages: it produces useful and meaningful variation in human responses to model that reveals graded category structure, whereas the majority of images are identifiable once upsampled. On the other hand, the kind of ambiguity induced by low-resolution images is akin to removing information (and introducing perceptual noise), which contrasts with the kind of ambiguity introduced in previous work employing highly similar stimuli such as oriented Gabor filters[29]. Finally, the dataset contains a reasonable number of borderline examples that are ambiguous between two or more categories (medium- and high-entropy guess distributions; Fig. 3c), in contrast to high-resolution datasets over hundreds of categories that are more carefully curated—more in keeping with the nature of experimental categories explored in previous work[17].

We find our CIFAR-10H dataset has a number of attractive properties. First, there is its size: roughly 50 categorizations from different subjects for each of 10,000 images from ten natural categories. Having this many judgments gives us the statistical power to fit the larger number of free parameters that are necessary to extend cognitive models to high-dimensional stimuli.

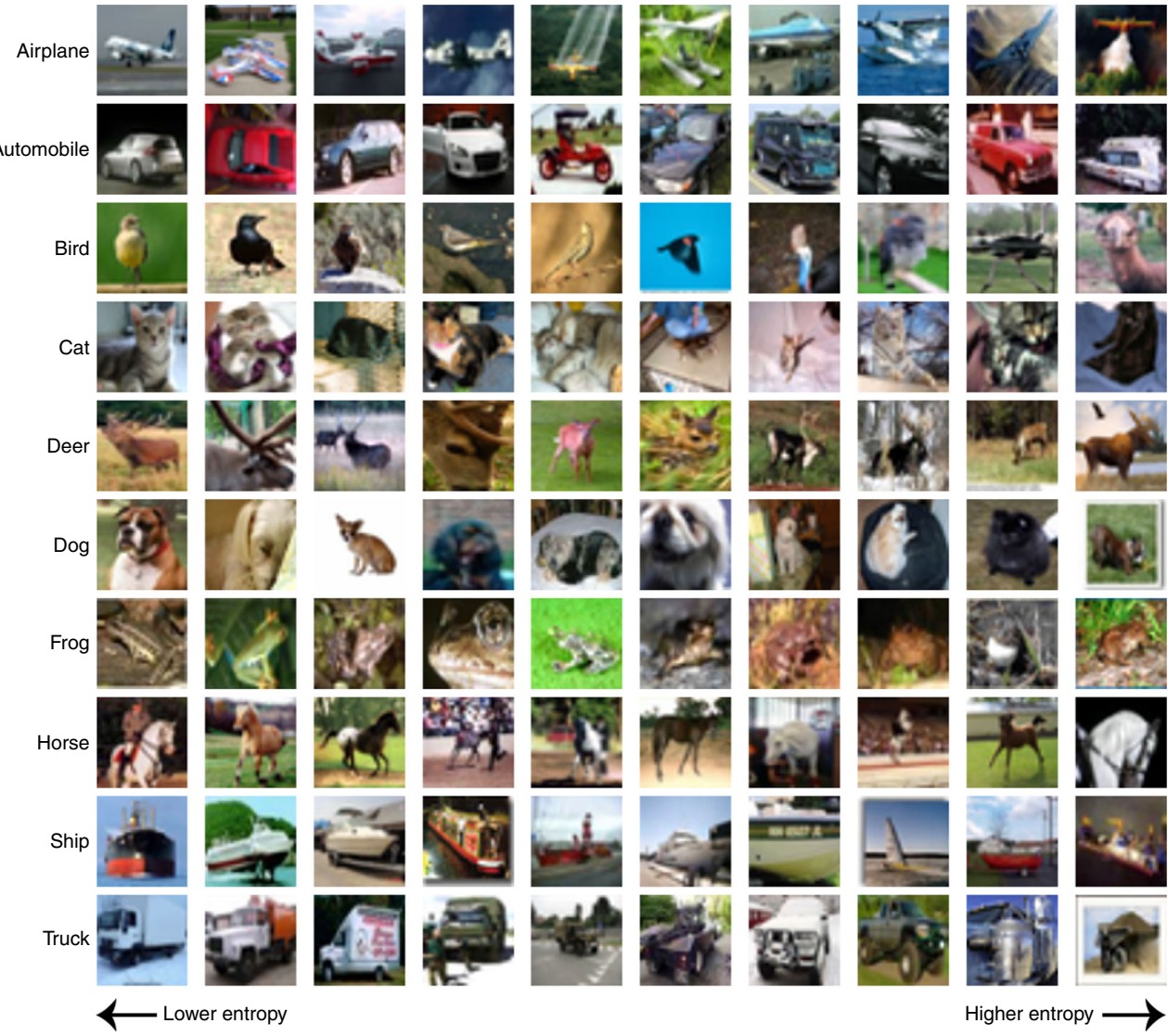

**Fig. 2 Stimuli for current study.** Ten representative natural images from each of the categories we use in our experiments, arranged by the entropy of their associated human categorizations. Images were taken from the `CIFAR-10` test set[49] and clustered into ten entropy bins, with the same minimum and maximum entropy used for all categories. Representative images were then chosen from each bin, for each category. Lower entropy means a higher degree of human categorization consensus.

These free parameters endow models with the flexibility to transform representations to better correspond to human psychological representations during the training phase (in other words, avoid underfitting)—a property that we below show improves our modeling of human confusions for ambiguous natural images. Conversely, we have enough datapoints so that these free parameters are constrained to avoid generating spurious solutions with these more flexible models (in other words, avoid overfitting).

Encouragingly, two features of the human responses to these images support their choice. First, the entropy distribution of these judgments roughly follows a power-law distribution, meaning that for most images there is a high degree of human consensus, but there are also a sufficient number of images in which human consensus diverges (see Fig. 3c). This property reflects our intuitions about human categorization, in which most stimuli are usually readily identifiable into one or two categories and have graded category membership[10]. In addition, the confusion structure we find parallels what we would intuitively

expect from the natural world: the most confused categories are dogs and cats, horses and deer, and automobiles and trucks (see Fig. 3d).

**Rich feature representations improve categorization models.** It seems intuitive that we categorize a novel stimulus based on its similarity to previously learned categories or memorized examples from them. This motivates a common framework for existing computational models: categorization as the assignment of a novel stimulus, $y$, to a category, $C$, based on some measure of similarity $S(\mathbf{y}, C)$ between a vector of feature values, $\mathbf{y}$, and those of existing category members in $C$. We can use the Luce-Shepard choice rule[53,54] to relate similarities to probabilities and determine the likelihood of a single categorization, made over our ten categories:

$$p(\text{Guess } Category \ i | \mathbf{y}) = \frac{S(\mathbf{y}, C_i)^{\gamma}}{S(\mathbf{y}, C_1)^{\gamma} + \ldots + S(\mathbf{y}, C_{10})^{\gamma}} , \quad (1)$$

where $\gamma$ is a freely estimated response-scaling parameter.

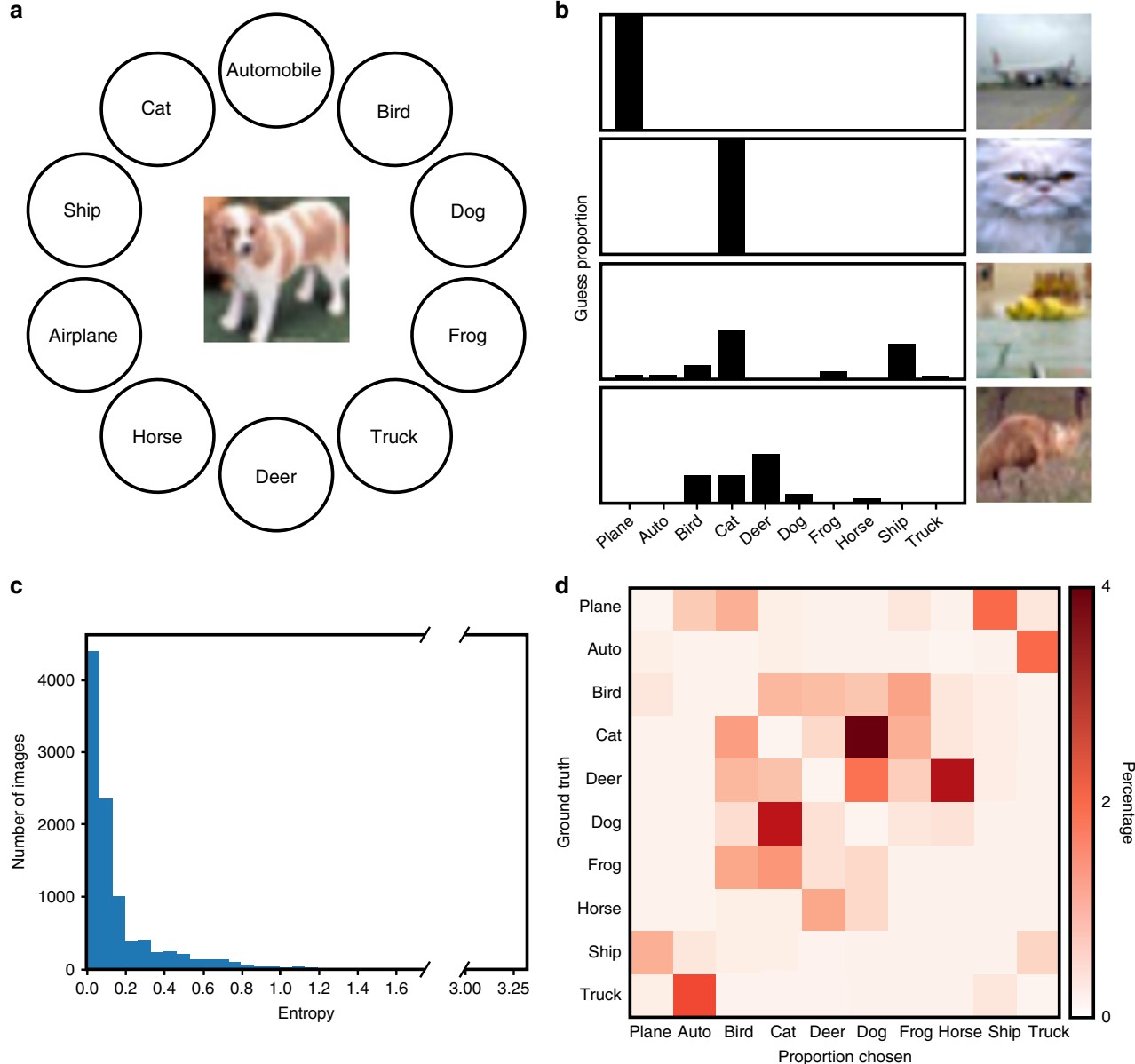

**Fig. 3 Task paradigm and behavioral data. a** Experiment web interface for our human categorization task. Participants categorized each image from an order-randomized circular array of the `CIFAR-10` labels. **b** Examples of images and their human guess proportions. For many images (upper plane and cat), choices are unambiguous, matching the `CIFAR-10` labels. For others (lower boat and bird), humans are far less certain. **b** The number of images by the entropy of their associated human guess distributions (less entropy means humans had greater consensus). **d** The confusion matrix for categories across all human judgments. Source data are provided as a Source Data file.

For both prototype and exemplar models, we use an exponentially decreasing function of distance in the stimulus feature space to measure the similarity between two stimulus vectors **y** and **z**[55]:

$$S(\mathbf{y},\ \mathbf{z}) = \exp\{-d(\mathbf{y}, \mathbf{z})\} \ . \tag{2}$$

The distance calculations for our models can be united under the framework of computing the Mahalanobis distance. This means that the distance between two stimuli is given by the following equation:

$$\mathrm{MHD}\ (\mathbf{y},\ \mathbf{z}) = (\mathbf{y} - \mathbf{z})^{T}\ \Sigma_{C_z}^{-1}\ (\mathbf{y} - \mathbf{z})\ , \tag{3}$$

where $\Sigma_{C_z}^{-1}$ is the inverse of the covariance matrix for the category that **z** belongs to. We take $\Sigma$ to be a diagonal matrix, so the distance reduces to computing $(y - z)^2/\sigma^2$ for each dimension,

where $\sigma^2$ is the diagonal entry (the variance) associated with that dimension. The parameters of $\Sigma$ can thus be thought of as reweighting the dimensions used for calculating distances: a standard component of categorization models[13] and exactly the transformation that was found to produce a close correspondence between CNN representations and human judgments in previous work[48].

In a prototype model[12], a category prototype—the average of category members—is used for this similarity calculation: $C$ is represented by the central tendency, $\boldsymbol{\mu}_C$, of the members of category $C$ and $S$ measures the similarity between a novel stimulus vector and this category prototype—this creates a simple decision boundary in stimulus representational space between category prototypes. We define three Prototype models: Classic, Linear, and Quadratic. The Classic model follows the traditional formulation of a prototype, assuming uniform variance for

**Table 1 Categorization model variants: descriptions and notation.**

| Model name | Parameter | Optimization | $N_p$ | $\Sigma_{C_z}^{-1}$ |
|---|---|---|---|---|
| Classic Prototype | Category prototypes: $\mu$ | Fixed | 0 | |
| | Category inverse variance: $\mathbf{I}$ | Fixed | 0 | $\mathbf{I}$ |
| | Choice rule: $\gamma$ | Estimated | 1 | |
| Linear Prototype | Category prototypes: $\mu$ | Fixed | 0 | |
| | Category inverse variance: $\mathbf{cI}$ | Estimated | $N_d$ | $\mathbf{cI}$ |
| | Choice rule: $\gamma$ | Estimated | 1 | |
| Quadratic Prototype | Category prototypes $\mu$ | Fixed | 0 | |
| | Category inverse variances: $\mathbf{c_iI}$ | Estimated | $N_c \times N_d$ | $\mathbf{c_zI}$ |
| | Choice rule: $\gamma$ | Estimated | 1 | |
| Exemplar (no attention) | Category exemplars | Fixed | 0 | |
| | Distance scaling: $\beta$ | Estimated | 1 | $\beta\mathbf{I}$ |
| | Choice rule: $\gamma$ | Estimated | 1 | |
| Exemplar (attention) | Category exemplars | Fixed | 0 | |
| | Attentional weights: $\mathbf{w}$ | Estimated | $N_d$ | $\beta\mathbf{wI}$ |
| | Distance scaling: $\beta$ | Estimated | 1 | |
| | Choice rule: $\gamma$ | Estimated | 1 | |

$N_c$ number of categories, $N_d$ number of feature dimensions, $N_p$ number of parameters, $\Sigma_{C_z}^{-1}$ inverse of category covariance matrix in Mahalanobis distance calculation.

all categories ($\Sigma_C = \sigma^2\mathbf{I}$ for all $C$, where $\sigma^2$ is a scalar constant and $\mathbf{I}$ is the identity matrix) and so reduces to calculating the Euclidean distance[12]. The Linear model assumes the categories share variance along each dimension ($\Sigma_C = \Sigma$ for all $C$, where $\Sigma$ has non-zero entries only on the diagonal) and the Quadratic assumes that each category has its own variance along each dimension (each $\Sigma_C$ is a distinct diagonal matrix), allowing them to define more complex decision boundaries (linear and quadratic, respectively)[16]. In an exemplar model[13], all memorized category members are used: $C$ is represented by all existing members or exemplars of category $C$ and $S$ measures the sum of similarities between the novel stimulus vector and all category exemplars, allowing the model to capture complex non-linear decision boundary between categories that, with sufficient samples, can approximate any Bayes optimal decision boundary[15]. It is in this function approximation sense, which is widely used in machine learning, that we consider exemplar models more complex than prototype models. We compare two well-known exemplar models: one that assumes uniform variance along each dimension and one with attentional weights that assumes all exemplars share the same variance along each dimension (the same assumption as the Linear Prototype model). We present the best-scoring of these as our Exemplar model (see Table 1 and "Methods" for details about how these assumptions about variance translate to free parameters). Finally, we present two baseline models for each of the CNN representations, which were created by using their output probabilities as similarities and inserting into the choice rule given above.

The formalization outlined above continues the tradition of modeling categorization as a probabilistic procedure, in which a novel stimulus, $\mathbf{y}$, is assigned to the category it is most similar to. We make a notational departure from this tradition in using the Mahalanobis distance to unite the similarity comparisons of prototype and exemplar models. We find this helpful for two reasons: first, because it allows a more intuitive description of our categorization models. In particular, we can see that in prototype and exemplar strategies, a Gaussian distribution is used to model categories and category exemplars, respectively, with the importance of a dimension weighted inversely to the variation along it. Under the minimal assumptions of learners having access to information on the location and variance of categories or exemplars, this equates to choosing the maximum entropy distribution to guide our comparison. In light of this, our set of models can be thought of as defining a range of boundary

complexities over stimulus space, rather than competing model classes with different metrics, as generally depicted in previous work. Second, it makes comparison of mathematical details between different model types from within these classes more clear: for instance, we see that our Linear Prototype and Exemplar model with attentional weights are performing essentially the same distance comparison, using a similar number of free parameters. Finally, this formulation allows us to extend these modeling strategies to a broader and more flexible class. We can see this with our Quadratic prototype model, which, although evaluated in earlier work[16], has been mostly absent from more recent studies. Although we could also define more complex exemplar models, this would be too computationally intensive for the current work—instead, we leave this direction open for future investigation.

We assess the performance of these categorization models using five types of stimulus representations from computer vision: unprocessed pixel data (pixels), Histograms of Oriented Gradients[56] (HOGs), latent space encodings from a bidirectional generative adversarial network[57] (BiGAN), and final-layer activations from two well-known CNN classifiers AlexNet[58] and DenseNet[59]. This set of representations coarsely recapitulates the history of natural image featurization, with raw pixel inputs being the obvious starting point. HOG features are slightly higher-level descriptors of local features that are not learned from data. AlexNet, which is in our case a CIFAR-sized variant of its namesake, spearheaded the popularity of deep learning and remains a simple yet effective deep CNN classifier. DenseNet represents some of the architectural progress made (in this case, skip connections) in the years since AlexNet. Finally, BiGAN is able to obtain high performance on classification tasks after adversarial training in which no category labels are used for representation learning (in other words, no supervision). This diverse set also allows us to independently factor out contributions of convolutional layers (AlexNet, DenseNet, and BiGAN) and supervised (AlexNet and DenseNet) *versus* unsupervised (BiGAN and HOG) learning, which are often conflated when discussing CNNs. AlexNet and DenseNet were pre-trained on the 50,000-image training subset of CIFAR-10 to a test-set classification accuracy of 82% (AlexNet) and 93% (DenseNet). Two-dimensional linear-discriminant-analysis plots of these representations are shown in Fig. 4.

When we stratify categorization model performance by feature type and model class, we find that the nature of the underlying

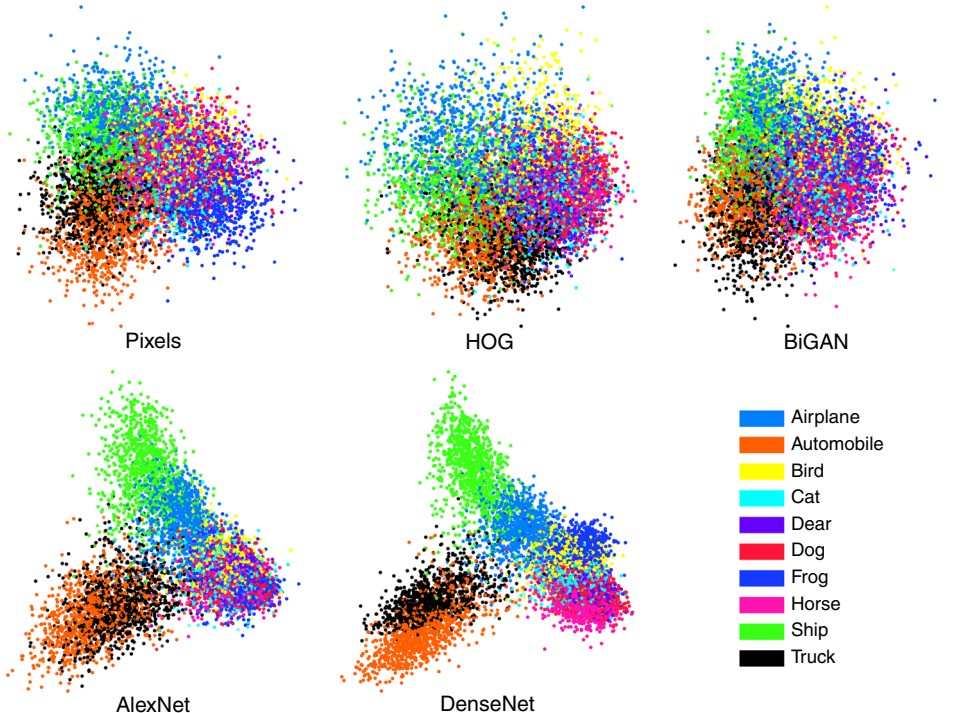

**Fig. 4 Stimulus representations.** Two-dimensional linear-discriminant-analysis projections of the five stimulus representations used in our analyses, for all stimuli (colored by category). As the representation quality increases from left to right, the apparent class separability increases while within-class variability decreases.

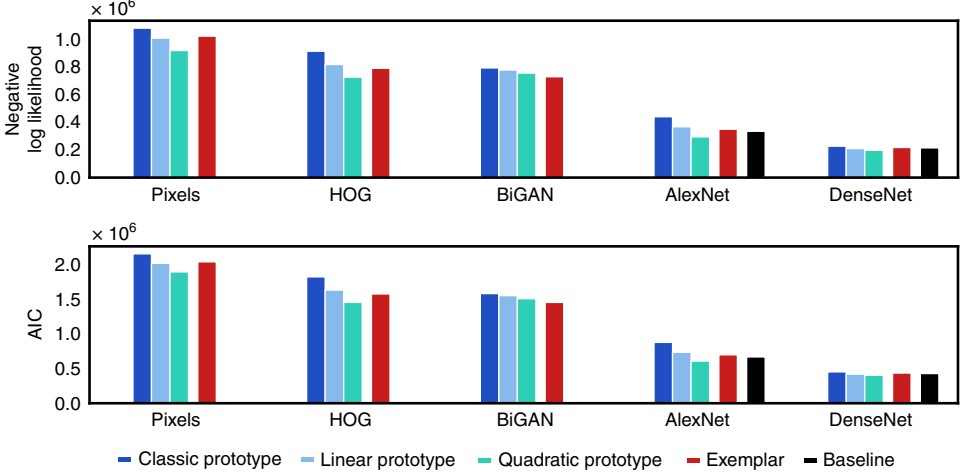

**Fig. 5 Categorization model results.** Negative log-likelihood and AIC results for all categorization models using all five stimulus representations (lower scores mean better performance). Within each representation, models are colored and clustered by class (prototype, exemplar). Baseline model performances are shown using a black bar for relevant feature spaces (AlexNet and DenseNet). Source data are provided as a Source Data file.

feature space greatly affects categorization model performance—much more than the class of categorization model used (Fig. 5). In particular, there appears to be a clear distinction between models based on simple (AlexNet) and more advanced (DenseNet) supervised CNN stimulus representations, which capture human decisions well, and those based on raw pixels (Pixels), hand-engineered computer vision features (HOG), and an unsupervised generative CNN (BiGAN), which capture human decisions poorly. This effect does not appear to be obviously related to the dimensionality of stimulus representations: HOG and DenseNet representations are roughly the same dimensionality, AlexNet has the second largest feature space, and BiGAN

representations are the smallest. Nor is it explained by the use of convolutional architectures: BiGAN representations, while able to support effective generative image modeling and comparable downstream object classification using convolutional layers in the original work[57], perform considerably worse than the much older supervised AlexNet CNN. It does, however, appear related to whether the representations come from networks that have been designed to solve natural image classification in a supervised manner (namely, AlexNet and DenseNet).

All parameter estimation was performed by cross-validation, making it unlikely that our models are overfit. However, as the log-likelihood improvements for our best-performing models

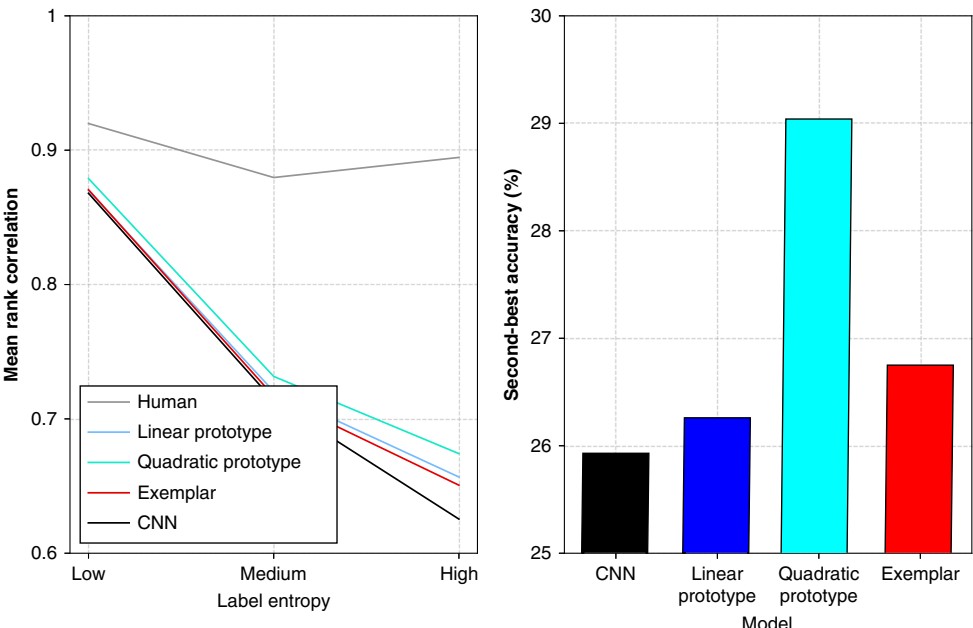

**Fig. 6 Correlation results and second-best accuracy.** Left: average Spearman's rank correlation coefficient for images with low-, medium-, and high-entropy guess distributions. Right: second-best accuracy (SBA) for best-performing CNN baseline and categorization models. Source data are provided as a Source Data file.

over the one-parameter Classic prototype model and CNN baselines are partly a product of increased expressivity, we present Akaike Information Criterion[60] (AIC) scores as an alternate measure of model fit that penalize more highly-parameterized models. There are few notable changes as a result, indicating the differences in results unlikely to be solely due to an increased number of free parameters.

**Cognitive models improve prediction for ambiguous images.** Next, we observe that although the baseline model performs very well for the DenseNet representations, the use of cognitive models still confers an improvement, capturing more of the fine-grained graded category structure present in human categorizations. This can be seen more easily if we stratify categorization model performance by the level of uncertainty that people collectively exhibit in judging images (Fig. 6 left). For images with high human consensus (and therefore low guess-distribution entropy), the CNN baseline and cognitive models perform comparably. This is in a sense unsurprising, given that for the majority of images there was a high human consensus on the single label the CNN is pre-trained to predict to a high accuracy. However, the Quadratic Prototype model and the Exemplar model still outperform this baseline model overall, and in particular model graded category structure better for more interesting images where humans were less certain (medium and high entropy).

Importantly, these improvements do not seem to be made at the cost of losing confidence for images where humans and the neural network baseline were certain, which is non-trivial given the complexity of the underlying representational space they must transform. The improvements to capturing human graded category structure can also be seen in the greater second-best accuracy (SBA) scores—the percentage of the time the model correctly predicted the second most common human choice—for these models, where, for example, the best-performing prototype model improves performance by 3% over the baseline and 2% over the Exemplar model over 10,000 images (Fig. 6 right). These successful models incorporate free parameters that can be used to modify the underlying features to better model human

categorization; the Classic Prototype model, by contrast, does not. Given this pattern of results, we can see that transforming supervised CNN representational spaces with cognitive categorization models makes it possible to model human categorization decisions well, and that the value of these cognitive models should increase with the underlying uncertainty of the dataset. In this context, it is interesting that although the DenseNet feature space is much more compressed than AlexNet—with roughly one-third the number of dimensions—it allows for more information relevant to the related task of human categorization to be retained.

**Prototype models perform comparably to exemplar models.** Our third major finding is that, for all representational spaces, there is a qualitatively different pattern of results than would be expected from previous experiments in lower-dimensional artificial spaces: prototype models (Linear and Quadratic) perform comparably to exemplar models (see Figs. 5 and 6). This result is surprising given the frequent superiority in fit by exemplar models in previous work, and the fact that prototype models form simple decision boundaries and only make a single distance comparison per category, whereas exemplar models can form arbitrarily complex boundaries and make $|C| = 1000$ comparisons (in our dataset).

The relative performance of prototype and exemplar models is determined by two factors: the dimensionality of the space and the structure of the categories in that space. To investigate the former, we simulated categorization model performance over increasingly complex artificial categories and varied the number of training samples and the dimensionality of stimulus representations (see Fig. 7). These simulations demonstrate that intuitions about model performance developed for low-dimensional categories do not directly transfer to higher-dimensional ones. In particular, for more complex categories (see Fig. 7, rightmost columns), we found that increasing the dimensionality of stimuli resulted in a difference in the relative performance of prototype and exemplar models. For complex categories over a lower number of dimensions (see Fig. 7,

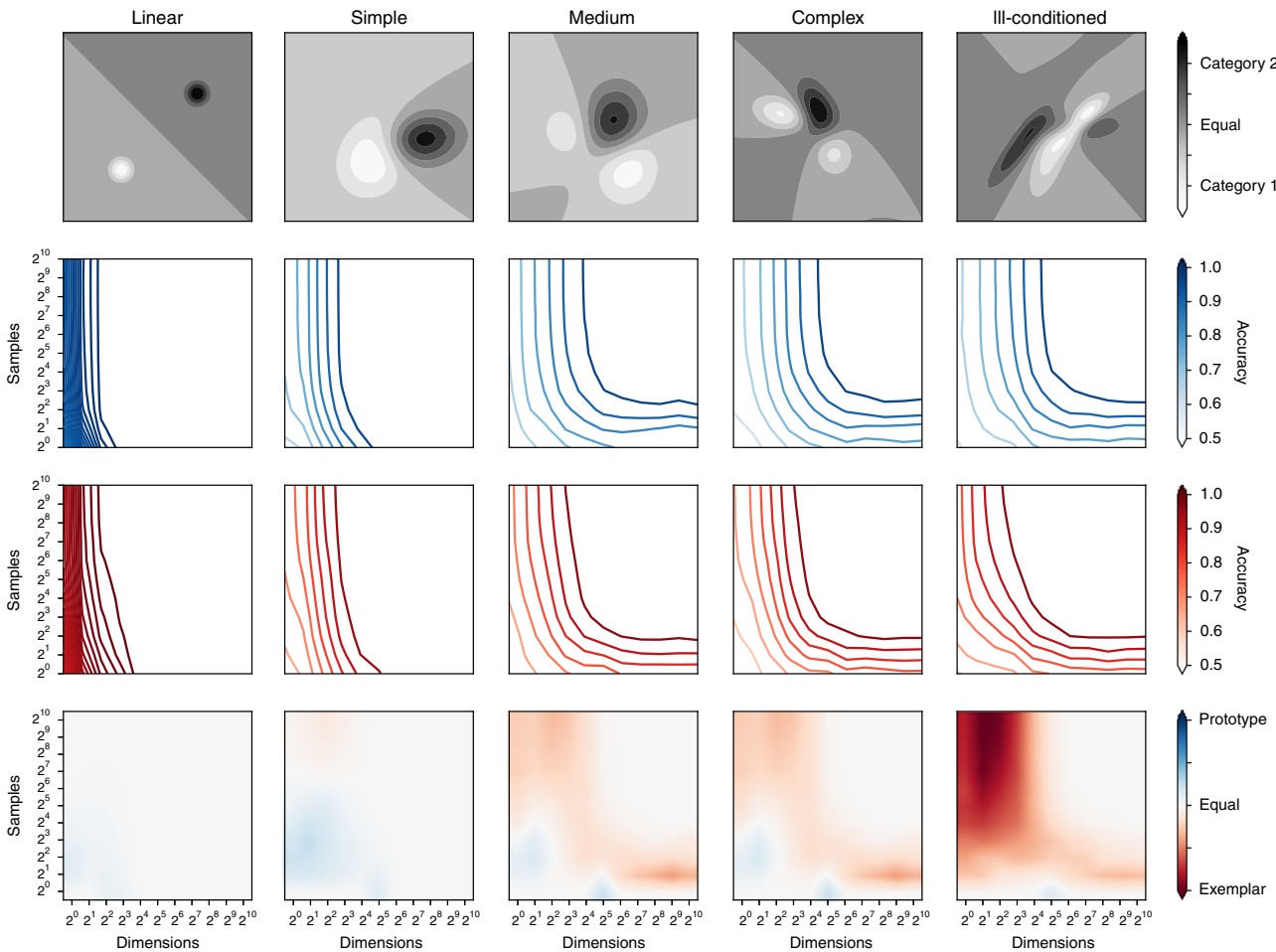

**Fig. 7 Performance of categorization models depends on dimensionality of stimuli.** Categories of increasing complexity were created (columns, left to right), and prototype and exemplar accuracy compared on stimuli sampled from them. Top row: representative category structures. Middle rows: prototype (blue) and exemplar (red) model accuracy, as the number of dimensions (horizontal axes) and training samples (vertical axes) varies. Bottom row: comparison of middle rows (prototype accuracy—exemplar). Each point represents the averaged results from 300 different distributions, with bilinear smoothing between points. In different regimes, the relative performance of prototype and exemplar models varies significantly, in particular between the regimes covered in laboratory studies (top left) and those necessary to cover more naturalistic categorization studies (top right). Source data are provided as a Source Data file.

rightmost columns, left side of plots), exemplar models increasingly outperform prototype models as the number of training samples increases, which describes most previous laboratory studies providing the strongest support for exemplar models[17,61]. However, for more complex simulated categories in higher-dimensional spaces (see Fig. 7, rightmost columns, right side of plots), both models perform equally well with many samples. This finding is consistent with the idea that the mathematical constraints on the form of category boundaries that both models can learn have different consequences, in terms of performance, in low- and high-dimensional spaces. For low-dimensional spaces, where there are fewer possible ways to separate categories, the strong constraints imposed by the functional form of prototype models—, namely, having a hyperplane as a decision boundary—means there are situations where the exemplar must perform better given enough samples. However, in high-dimensional spaces these constraints do not impose the same penalty: prototype models can perform as well as exemplar models. If the spatial assumptions underlying all modern categorization models continue to be employed, it is highly likely that we will need many dimensions to mathematically model categorization over a large and varied range of stimuli that vary in natural ways. What these simulation results show is that there is a

theoretical consequence of using those dimensions that interferes with intuition and the extension of results developed in experimental settings using simple artificial stimuli. It also helps explain the results shown in Fig. 5, in which for the number of feature dimensions that current models require to classify natural images well (roughly 64–1024), the simple decision boundaries formed by prototype models appear to perform well in comparison to exemplar models.

If the categories are structured in a way that is consistent with prototypes (namely that category members are relatively well clustered and categories can be separated with simple boundaries), then prototype models will do well regardless of dimensionality, as illustrated in the leftmost panels of Fig. 7. Any conclusions based on the representations from AlexNet and DenseNet need to be tempered by the fact that these representations were explicitly constructed to allow the neural networks to classify stimuli via linear boundaries in these spaces. As a consequence, we might expect that prototype models would do well in predicting human categorization judgments using these representations. However, we note that the Linear and Quadratic prototype models also perform comparably to the Exemplar model using pixels or HOG features, which are not designed to support linear boundaries but do have very high dimensionality.

## Discussion

It is testament to the importance of categorization in accounts of cognition that it remains a focus of research and discovery after so many years of investigation. Continuing this rich tradition, in the present study we have asked whether we might be able to extend theory matured in simplified laboratory settings to capture human behavior over stimuli more representative of the complex visual world we have evolved and learned within. Previous work on formal models of categorization had at least two motivations for focusing on low-dimensional, artificial stimuli. First, these formal models require stimuli to be represented in a feature space, yet the psychological feature spaces that define natural categories are unknown. This means a potentially large representational gap exists between the inaccessible mental representations of stimuli and our mathematical approximations of them, which can be ameliorated by defining obvious dimensions of variation for artificial stimuli (see Fig. 1). For more naturalistic stimuli, however, it is unlikely their variation can be suitably reduced or attributed to a few dimensions, or category boundaries learned over such a compressed space. Any attempt to model categorization over such stimuli must therefore address the problem of finding appropriate representations for large numbers of varied naturalistic stimuli. Second, armed with simple stimuli, experimenters have been able to control the members of a category that an individual experienced, and so have some degree of confidence that the categorization models they were running reflected the true psychological models. Clearly, this sort of control can never be matched with most naturalistic stimuli. Any successful extension of prior formal models to naturalistic stimuli must directly address these concerns.

In the work presented above, we have adapted the traditional methodological approach with these aims in mind. First, we use a large, diverse collection of natural images as stimuli. Second, we use state-of-the-art methods from computer vision to estimate the structure of these stimuli, which our models further adapt based on behavioral training data. This contrasts with the majority of previous work, in which a small number of a priori-identified features were manipulated to define and differentiate categories, and as a consequence were limited to simple artificial stimuli. Finally, we offset the uncertainty these advances introduce by using a large behavioral dataset to more finely assess graded category membership over stimuli. Taken together, our results show that using representations derived from CNNs makes it possible to apply psychological models of categorization to complex naturalistic stimuli, and that the resultant models allow for precise predictions about complex human behavior. Our most general finding is that categorization models that incorporate supervised CNN representations predict human categorization of natural images well—in particular, exemplar and prototype models that are able to modify CNN representations using information about human behavior. This is theoretically interesting because it indicates there is enough latent information and flexibility in the existing CNN representations to harness for such a related task, and that cognitive models can exploit it to derive graded category structure.

Working with these complex, naturalistic stimuli reveals a more nuanced view of human categorization. The broad consensus from decades of laboratory studies using simple artificial stimuli was that people could learn complex category boundaries of a kind that could only be captured by an exemplar model[17]. Extrapolating from these results, we might imagine that categorization should be thought of in terms of learning complex category boundaries in simple feature-based representations. Our results outline an alternative perspective. We see from our simulations that when operating in the type of high-dimensional spaces necessary to represent large numbers of varied naturalistic stimuli, the relative ability of prototype and exemplar models depends on dimensionality, training set size, and model error. Furthermore, the nature of these feature spaces themselves appears to affect categorization profoundly, to the extent that the choice of feature space would seem as important as the choice of categorization strategy. When thinking about humans, feature representations are likely to have been learned early on, through a slow, data-driven learning process. Given these considerations, one might expect psychological representations to reflect the natural world, such that categorization of natural stimuli is made as efficient and as simple as possible; of the type that could be easily classified by a prototype-like model. On the other hand, artificial or unlikely stimuli may at times carve out awkward boundaries in these spaces, which perhaps also underlies the equivocal performance of exemplar models in our work. What seems most clear is that incorporating feature learning into studies of human categorization, as has been called for in the past[62], and developing a deeper understanding of how these processes interact in the context of complex natural stimuli is an important next step towards more fully characterizing human categorization. For such an analysis, past theoretical and simulation studies exploring the conditions in which prototype and exemplar models do and do not mimic each other are likely to be of great relevance[16,63,64].

We view our approach as complementary to recent work training CNNs to predict the MDS coordinates of naturalistic images and using these as input for categorization models[43]. The key difference is the source of stimulus representations, their generality, and their cost to procure. MDS-coordinate predictions, if reasonably approximated, capture a great deal of information about the aspects of stimuli that are psychologically meaningful. However, in order to train networks to predict them, human similarity judgments between images must be collected—an expensive task, as the number of pairwise similarity judgments grows quadratically with the number of images. Equally importantly, it remains to be shown that CNNs trained to predict MDS solutions for small sets of images will generalize well to more varied stimulus sets. On the other hand, CNNs are routinely trained to classify thousands and even millions of natural images with high accuracy, indicating that the information in their representations is somewhat consistent for a wide range of complex visual stimuli. Using these direct estimates of stimulus structure could be considered a less precise approximation of human mental representations; however, the already highly relevant information they encode is easily improved by the simple transformations implicit in almost all of our top-performing models, which only require a training set of categorization data that grows linearly with the number of images. Interestingly, our simulation and results imply it is likely that as the number of dimensions of variation climbs in a dataset, the categorization results generated by both methods will begin to converge and so we look forward to a test of this theory with future work.

The use and adaptation of machine learning techniques and representations to extend psychological research into more naturalistic domains is a field in its infancy. However, as we simply cannot access human psychological representations for such complex naturalistic stimuli, the preliminary success of our approach is encouraging, and we are likely to see further benefits that track the progress of large image databases and improvements in deep network architectures. One natural extension here would be to investigate the feasibility of using a machine learning dataset with lower-level category labels; for example, a subset of the ImageNet database[58]. Beyond their interest for psychology, human behavioral data for these domains provides a rich yet largely unexploited training signal for computer vision systems, as we recently demonstrated by using such information to improve

the generalization performance and robustness of natural image classifiers[65]. More broadly, these results highlight the potential of a new paradigm for psychological research that draws on the increasingly abundant datasets, machine learning tools, and behavioral data available online, rather than procuring them for individual experiments at heavy computational and experimental cost. Towards this aim, the large dataset we offer in this work can provide a direct bridge between frontline efforts in machine learning and ecologically valid cognitive modeling, the two of which we hope continue to develop in synergy.

## Methods

**Stimuli**. Our image stimuli were taken from the CIFAR-10 dataset, which comprises sixty thousand $32 \times 32$-pixel color images from ten categories of natural objects[49]. We collected human judgments for all 10,000 images in the "test" subset, which contains 1000 images for each of the following 10 categories: airplane, automobile, bird, cat, deer, dog, frog, horse, ship, and truck. For the web-based experiment, we upsampled these images to $160 \times 160$ pixels using scipy's "bicubic" image upsampling function[66].

We chose to use images from this dataset for a number of reasons. First, the dataset has a long and still active history of exploration by the machine learning community, meaning that in addition to the range of representations already available, it is likely that the baseline representations and any innovations from the present work will continue to improve the fit to humans. Second, the number of images is still small enough for us to collect enough human judgments to offer a good approximation of the underlying population-level guess distribution for the entire test set. Third, the low resolution of the images is actually advantageous: it produces useful yet meaningful variation in human responses that reveals graded category structure, whereas the majority of images are identifiable once upsampled (see Fig. 3). Finally, the dataset contains a reasonable number of borderline examples that are ambiguous between two or more categories (medium- and high-entropy guess distributions), in contrast to high-resolution datasets over hundreds of categories that are more carefully curated—more in keeping with the nature of experimental categories explored in previous work[17].

**Human behavioral data**. Our CIFAR-10H behavioral dataset consists of 511,400 human categorization decisions made over our stimulus set collected via Amazon Mechanical Turk. In our large-scale experiment, 2570 participants were shown upsampled $160 \times 160$-pixel images one at a time and were asked to categorize it by clicking one of the 10 labels surrounding the image as quickly and accurately as possible (see Fig. 3a). Label positions were shuffled between participants. There was an initial training phase, during which participants had to score at least 75% accuracy, split into 3 blocks of 20 images taken from the CIFAR-10 training set (60 total, 6 per category). If participants failed any practice block, they were asked to redo it until passing the threshold accuracy. After a successful practice, each participant categorized 200 images (20 from each category) for the main experiment phase. After every 20 trials, there was an attention check trial using a carefully selected unambiguous member of a particular category. Participants who scored below 75% on these checks were removed from the final analysis (14 participants failed checks). The mean number of judgments per image was 51 (range: 47–63). The mean accuracy per participant was 95% (range: 71%–100%). The mean accuracy per image was 95% (range: 0%–100%). Average completion time was 15 minutes, and all subjects were compensated for their work (each participant was paid $1.50). Informed consent was collected from all subjects, while the authors were at the University of California, Berkeley, with ethical approval given to the study protocol by the Institutional Review Board under protocol number 2015-05-7551. We complied with all relevant ethical regulations.

We collected this amount of data for two reasons. The first is that, given preliminary work, we knew roughly the number of categorizations needed to well-approximate the human uncertainty over images. The second is that to fit categorization models (for example, the Linear and Quadratic Prototype models, and the Exemplar model with attentional weights) properly in the high-dimensional feature spaces learned by contemporary machine learning classification models, a certain number of parameters are needed to establish models with the flexibility to transform representations to fit human behaviour during the training phase (that is, to avoid underfitting) and a certain number of datapoints are needed to avoid generating spurious solutions with these more flexible models (that is, to avoid overfitting). The number we collect allows us to fit (roughly) several vectors per category (covering the Linear and Quadratic Prototype models and the Exemplar model with attentional weights).

In Fig. 2, we use these category judgments to help visualize our dataset. First, we found the minimum (0.0) and maximum entropy common to all category guess distributions. Then, we divided this scale into ten increasing bins, and sorted images into these bins based on the entropy of their guess distributions. Finally, we selected one representative image from each category for each entropy bin, and displayed these so that they were aligned across categories.

**Categorization models**. Formally, we can unite prototype and exemplar strategies of categorization under the following common framework: categorization as the assignment of a novel stimulus $\mathbf{y}$ to a category $C$ based on some measure of similarity $S(\mathbf{y}, t)$ between feature vector $\mathbf{y}$ and those of existing category members (expressed in a summary statistic $t_C \equiv f(\mathbf{x} : \mathbf{x} \in C)$). This allows us to fully specify a model by a summary statistic $t$, a similarity function $S$, and a function that links similarity scores for each category to the probability of selecting that category given $\mathbf{y}$.

The summary statistic $t$ describes the form of the category description that can vary under different categorization strategies. In a prototype model[12], a category prototype—the average of category members—is used for comparison: $t_C$ becomes the central tendency $\boldsymbol{\mu}_C$ of the members of category $C$. In an exemplar model[13], all memorized category members are used: $t_C$ represents all existing members or exemplars of category $C$ and $\mathbf{y}$ is compared to all of them.

We take as our similarity function $S$ a standard exponentially decreasing function of distance in the stimulus feature space[55]. We also take $S$ to be an additive function: if $t$ is a vector, $S$ becomes the summation of the similarities between $\mathbf{y}$ and each element of $t$. Rearranging the formula assigning probability to guesses given above, we can reduce the probability of each judgment as follows:

$$p(\text{Guess Category } i \,|\, \mathbf{y}) = \frac{1}{\exp\left\{ \gamma \log\left( \frac{S(\mathbf{y}, t_{C_1})}{S(\mathbf{y}, t_{C_i})} \right) \right\} + \cdots + \exp\left\{ \gamma \log\left( \frac{S(\mathbf{y}, t_{C_{10}})}{S(\mathbf{y}, t_{C_i})} \right) \right\}}. \tag{4}$$

This defines a sigmoid function around the classification boundary, where $\gamma$ controls its slope, and therefore degree of determinism. As $\gamma \to \infty$, the function becomes deterministic, and as $\gamma \to 0$, it reduces to random responding. When formulated in this manner, the prototype model is equivalent to a multivariate Gaussian classifier[15] and the exemplar model to a soft-nearest-neighbors classifier. To evaluate the predictions of these models against human data, we record the category label, $C_i$, that the participant gives to the stimulus $\mathbf{y}_i$. We then compute the log likelihood of the $N$ human guesses under the model:

$$\mathcal{L} = \sum_{i=1}^{N} \log \frac{1}{\exp\left\{ \gamma \log\left( \frac{S(\mathbf{y}_i, t_{C_1})}{S(\mathbf{y}_i, t_{C_i})} \right) \right\} + \cdots + \exp\left\{ \gamma \log\left( \frac{S(\mathbf{y}_i, t_{C_{10}})}{S(\mathbf{y}_i, t_{C_i})} \right) \right\}}. \tag{5}$$

Finally, we show that this formalization of prototype and exemplar models can unified by using the Mahalanobis distance metric to compute the distance between vectors, subject to model-specific constraints (see subsections below). Prototype and exemplar models differ in how their similarity to a category $S(\mathbf{y}, t_C)$ is calculated, to which we now turn (see Table 1 for summary).

Prototype models can be shown to be equivalent to decision-bound models[16,67], and there is a formal correspondence between such models in the psychological literature and a particular subset of multivariate Gaussian classifiers from statistics in which the covariance of the Gaussian distribution describing each category is equal to the identity matrix[15]. For prototype models, similarity to a category is taken to be an exponentially decreasing function of the distance between a stimulus vector $\mathbf{y}$ and the category prototype:

$$S(\mathbf{y}, t_C) = \exp\{-d_C(\mathbf{y})\}. \tag{6}$$

A comparison between two categories can be expressed as a simplified ratio:

$$\frac{S(\mathbf{y}, t_{C_j})}{S(\mathbf{y}, t_{C_i})} = \frac{\exp\left\{-d_{C_j}(\mathbf{y})\right\}}{\exp\left\{-d_{C_i}(\mathbf{y})\right\}} = \exp\left\{-[d_{C_j}(\mathbf{y}) - d_{C_i}(\mathbf{y})]\right\}. \tag{7}$$

Classic prototype models use the (squared) Euclidean distance and when the prototypes are estimated empirically by averaging the whole set of ground-truth category members, we call this a Classic Prototype model.

We can extend the prototype class to include several more variants by recognizing that the squared Euclidean distance is a special case of the Mahalanobis distance metric:

$$\text{SQED}(\mathbf{y}, C) = (\mathbf{y} - \boldsymbol{\mu}_C)^T \, \mathbf{I} \, (\mathbf{y} - \boldsymbol{\mu}_C) \tag{8}$$

$$\text{MHD}(\mathbf{y}, C) = (\mathbf{y} - \boldsymbol{\mu}_C)^T \, \boldsymbol{\Sigma}_C^{-1} \, (\mathbf{y} - \boldsymbol{\mu}_C) \tag{9}$$

where $\boldsymbol{\mu}_C$ and $\boldsymbol{\Sigma}_C$ are the mean—or, prototype—and covariance matrix of category $C$.

If $\boldsymbol{\Sigma}_C$ is the same for all categories, then the decision boundary between competing prototypes in feature space is closest to is a hyperplane, resulting in a linear model[15]. Taking the empirical mean of ground-truth category representations as the prototype, $\boldsymbol{\mu}_C$, we can define a Linear Prototype model by also learning a single diagonal inverse covariance matrix common to all classes $C_i$—that is, $\boldsymbol{\Sigma}_{C_i}^{-1} = \text{diag}(\mathbf{c})$, where $\mathbf{c}$ is a vector fitted on training set data across all categories. If $\boldsymbol{\Sigma}_C$ is allowed to vary across categories, then this classification boundary can take more complex non-linear forms[15]. Again, by taking the empirical mean of ground-truth category representations as the prototype, $\boldsymbol{\mu}_C$, we can define a Quadratic Prototype model by also learning a diagonal inverse covariance matrix for each category $C_i$—that is, $\boldsymbol{\Sigma}_{C_i}^{-1} = \text{diag}(\mathbf{c}_i)$, where $\mathbf{c}_i$ is a vector fitted on training set data for each category $C_i$. Using a diagonal covariance matrix is equivalent to learning a weight for each dimension, a linear transformation that has previously been found to be sufficient to produce a close

correspondence between CNN representations and human similarity judgments[48]. Although there is the potential to learn the non-diagonal terms in these matrices, we do not have enough behavioral data to fit these without incorporating a more complex regularization framework to address overfitting. We also tested models in which we learned the category prototypes in the form of free parameters, instead of calculating them directly from the ground-truth representations. As these models scored roughly the same with many more parameters, we excluded them from further analysis.

Exemplar models compute the similarity between $\mathbf{y}$ and $t_C$ by taking the sum of the similarities between $\mathbf{y}$ and each known category member:

$$S(\mathbf{y}, t_C) = \sum_{\mathbf{x} \in C} \exp\{-\beta\, d(\mathbf{y}, \mathbf{x})^q\}\,, \tag{10}$$

where $q$ is a shape parameter (normally chosen to equal $r$, below) and $\beta$ is a specificity parameter[13]. The distance between two vectors is given by the following equation:

$$d(\mathbf{y}, \mathbf{x}) = \left[\sum_k w_k |x_k - y_k|^r\right]^{1/r}\,, \tag{11}$$

where we use $r = 2$, consistent with the use of integral dimensions that cannot be separately evaluated by participants. These models weights, $w_k$, are positive, sum to one, and are known as attentional weights. They serve to modify the importance of particular dimensions of the input. If these weights are fixed uniformly in advance —and in combination with the standard choice of $q = r = 2$ above—we obtain an exemplar model that uses a scaled squared Euclidean distance to compare stimulus vectors. If instead we allow attention weights to vary, we obtain a more flexible model. In the following analysis, we present the best performance of these models as our Exemplar model.

Computing the above equation with these hyperparameter settings corresponds exactly to solving the Mahalanobis distance between a novel stimulus vector and each of the existing category members under a particular set of constraints. To see this, note that with the choice of $q = r = 2$, the similarity comparison becomes the sum of squared Mahalanobis distances, with each distance comparison given by the following form:

$$\exp\{-\beta\, d(\mathbf{y}, \mathbf{x})^2\} = \exp\left\{-\beta \sum_k w_k (x_k - y_k)^2\right\} \tag{12}$$

$$= \exp\left\{-\beta\, (\mathbf{x} - \mathbf{y})^T\, \mathbf{w}\mathbf{I}\, (\mathbf{x} - \mathbf{y})\right\} \tag{13}$$

$$= \exp\left\{-(\mathbf{x} - \mathbf{y})^T\, \beta\mathbf{w}\mathbf{I}\, (\mathbf{x} - \mathbf{y})\right\}\,, \tag{14}$$

where $\mathbf{w}$ and $\beta$ are constrained as above, and $\mathbf{w}\mathbf{I}$ can be thought of as a diagonal inverse covariance matrix, $\Sigma_{C_i}^{-1}$, which is shared between categories (as in the Linear Prototype model detailed above). This means we can unite our prototype and exemplar model classes under the same mathematical evaluation. Finally, we note that because these models learn the $\beta$ parameter, they are free to unite and choose among interpretations of the exemplar model in which the best stored exemplar is used (corresponding to $\beta \to \infty$, or where every exemplar is weighted equally $\beta \to 0$, or anywhere in between).

**Stimulus representations**. To evaluate categorization models on our dataset, we need feature representations for each of our stimuli. We assess the suitability of a range of stimulus representations from contemporary computer vision models for this purpose, beginning with those from deep CNNs. Deep CNNs learn a series of translation-invariant feature transformations of pixel-level input images that are passed to a linear classification layer in order to classify large sets of natural images[44]. After training, a network will generate node activations at each layer for each image, forming vector representations that are increasingly abstract, and can be directly input into downstream statistical models. We extract feature representations for each of our stimuli from two popular off-the-shelf CNNs pre-trained on the training subset of the `CIFAR-10` dataset, which comprises 5000 images from each of the 10 categories listed above. Our first network was a version of AlexNet for `CIFAR-10`[58] that obtains a top-1 classification accuracy of 82% on `CIFAR-10`'s test subset using `Caffe`[68]. We use this network, because it has a simple architecture that allows for easier exploration of layers while maintaining classification accuracy in the ballpark of much larger, state-of-the-art variants. Our second network was a DenseNet with 40 layers, 3 dense blocks, and a growth rate parameter of 12, trained using the `keras` Python library to a top-1 accuracy of 93%[69]. We chose this network because it achieves near state-of-the-art performance while still being parameter efficient (for faster training). The key change DenseNet makes to the fundamental CNN structure is that each layer passes its output to all layers above (and as such creates a dense feed-forward connection graph), rather than just the subsequent layer, allowing for deeper networks that avoid vanishing gradients[59].

We also include three further sets of image representations for comparison. The first are the raw unprocessed pixel representations for each image, downloaded directly from the `CIFAR-10` website[49]. The second are HOGs for each image[56],

constructed using the Python `opencv` library[70]. These features are extracted without supervision and supported numerous computer vision tasks prior to modern CNNs. Our most successful HOG representation for which we report results used a window size of $8 \times 8$. The third set of representations are from the latent space of a BiGAN[57]. This network is convolutional, but unsupervised, and allows us to factor out the contribution of human-derived labels in training CNNs to learn representations useful for modeling people. Two-dimensional linear-discriminant-analysis plots of these representations are shown in Fig. 4.

**Training and evaluation**. We estimated all categorization model parameters with 5-fold cross-validation and early stopping using the Adam variant of stochastic gradient descent[71] and a batch size of 256 images (see Table 1 for exact parameters estimated for each model). We do not train the classification model parameters, including both CNNs: these are held fixed after pretraining on the `CIFAR-10` training dataset (see above). For each model, we conducted a grid search over the learning and decay rate hyperparameters for Adam, selecting the final model parameter set and early stopping point during training based on which gave the lowest cross-validated average log likelihood. We used the `python` package `Theano`[72] for categorization model specification, optimization, and analysis.

**Model comparison**. Our primary evaluation measure for each model was log likelihood. For all models, we computed these scores by generating predictions for all images in our stimulus set using the averaged cross-validated parameters taken at the early stopping point described above. We also use the AIC score to compare models, as it gives a score for each model that takes into account the relationship between the number of parameters they employ, and their log-likelihood scores[60]

$$\text{AIC} = 2k \;-\; 2\ln(\hat{\mathbf{L}}) \tag{15}$$

where $k$ is the number of parameters in the model and $\hat{\mathbf{L}}$ is the maximum log likelihood.

In addition to log likelihood, we use two more interpretable measures of how well our models capture graded category structure defined by the human guess distributions. The first is the average Spearman's rank correlation coefficient over images (Fig. 6 left). This reflects the average of how well our models captured the overall order of labels for each image, as derived from the human guess distributions. It is rare, however, that humans allocate some guesses to all categories for an image: in fact, the majority of images have guesses clustered in one or two categories, and seldom beyond five even in the high-entropy cases. Our probabilistic categorization models, by contrast, always allocate probability to every class for an image, even if these are vanishingly small. To allow us to only focus on the orderings between human guesses and significant model guesses, we round any model probabilities below the resolution of the human data (0.04; two guesses out of 50) to zero. This means large ordering discrepancies imposed by insignificant categorization model probabilities do not obscure the meaningful correlation. Finally, we stratify images based on their entropy by calculating the range in entropy of smoothed human guess distributions (using plus-one smoothing), and breaking this range into three bins. The first bin (low entropy: 0–0.64) contained around 8800 images, where each image had a high human consensus. The second bin (medium entropy: 0.64–1) had around 1000 images, and the third bin (high entropy: 1–1.37) around 200 images. In total, around 34% of our images had perfect human consensus for a single category. The second measure is SBA (Fig. 6 right). This measures whether the second most likely category label chosen by our models for an image is the same as by our participants. As both of these measures are averaged over thousands of images, the error bars are negligible and we have not included them.

**Baselines**. As baselines, we use the raw output (softmax) probabilities of both AlexNet and DenseNet neural networks for each image to provide $S(\mathbf{y}, t_C)$. In deep CNN classifiers, the softmax function takes the inner product between a matrix of learned weights and the rasterized output of the final pooling layer, and returns a probability distribution over all of the `CIFAR-10H` classes. These weights are learned based on minimizing classification loss over the training subset of the `CIFAR-10` dataset described above. Although not explicitly trained to output human classification probabilities, these models are the most competitive systems available in terms of making accurate classifications at the level of human performance. For this reason, and because the models output a full probability distribution over classes that may exhibit human-like confusion patterns, we expect them to provide a meaningful and competitive baseline with which to compare our model scores.

**Noise ceiling**. For the correlation analysis, we estimated the noise ceiling in human guesses by using a split-half method. We randomized human guesses for each image, and used half of these to predict the held-out half (using Spearman's rank correlation with the Spearman–Brown correction[73,74]). For each image, we averaged this correlation across 100 split-half models.

**Simulations**. Several key results from previous modeling work show that if artificial stimuli are sampled from overlapping—Gaussian—categories, they define

more complex optimal decision boundaries of the type that can be captured by non-parametric exemplar models but not parametric prototype ones[16,17]. Through the simulation experiments presented in Fig. 7, we wanted to assess how well this conclusion would generalize to higher-dimensional stimuli and categories.

To do this, we tested classic formulations of prototype and exemplar models on stimuli from five increasingly complex types of synthetic category over a range of dimensions. Category complexity was defined by varying the number of Gaussian distributions used to instantiate a category, and the parameters of those Gaussian distributions[17]. For the linear and simple case, our categories were single Gaussians. For the medium, complex, and ill-conditioned cases, the categories were mixtures of two Gaussians. Across these different types of category, we analyzed the predictions of a zero-parameter Classic prototype model and a one-parameter Exemplar model in which only the $\beta$ parameter was estimated (using a cross-validated grid search). All other model parameters were set to 1. Within each category type and dimension, we sampled a varying number of training stimuli to use to estimate the prototype and exemplar locations and then tested the models on a fixed number of unseen test stimuli (100 stimuli per category). We created 300 distributions for each type of distribution at each dimensionality and generated training samples randomly from them. Each point on the accuracy and comparison plots of Fig. 7 is the average from model results over these 300 distributions. Finally, we used the `python` package `Matplotlib`[75] with the bilinear interpolation method to create the plots themselves.

More formally, we define a category component using a Gaussian density as follows:

$$\mathbf{x} \sim N(\boldsymbol{\mu}, \Sigma_c), \tag{16}$$

$$\boldsymbol{\mu} \sim N(\mathbf{0}, \mathbf{I}), \tag{17}$$

$$\Sigma_c \sim \frac{1}{d}\text{Wishart}(d\mathbf{I}, d), \tag{18}$$

where $d$ is the degrees of freedom. In the linear category case, one component was used per category and $\Sigma_c$ set to $\mathbf{I}$ (isotropic/spherical), then shifted so that the means were far apart, creating a linearly separable dataset. For the simple case, one component was used per category, with $d \leftarrow N_d + 20$, with $N_d$ equal to the number of dimensions, as this was found to give two-dimensional categories that matched the simple categories used in seminal previous work[17]. For the medium case, each category was an even mixture of two Gaussian components, with $d \leftarrow N_d + 20$ and the closest pair of components defined as the first category. For the complex case, the same parameters were used except components were randomly paired to form categories. We found that examples from these two regimes matched the complex categories from the same study[17]. Finally, we note that setting $d \leftarrow N_d$ results in an ill-conditioned covariance matrix, and that categories constructed from randomly paired components of this nature had a more complex structure, often with high variance in a subset of dimensions. We call this type of distribution ill-conditioned.

**Reporting summary**. Further information on research design is available in the Nature Research Reporting Summary linked to this article.

## Data availability

The dataset presented in the current work (`CIFAR-10H`) is available in a public GitHub repository: https://github.com/jcpeterson/cifar-10h (https://doi.org/10.5281/zenodo.4008585). The image dataset (`CIFAR-10`[49]) can be found at https://www.cs.toronto.edu/kriz/cifar.html. Source data are provided with this paper.

## Code availability

All computer code is available on request from the authors.

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

## Acknowledgements

This work was carried out under the grant number 1718550 from the National Science Foundation. This grant was awarded based on the broader research program this work was conceived under and its funders had no role in the conceptualization, design, data collection, analysis, decision to publish, or preparation of the manuscript.

## Author contributions

R.M.B. and J.C.P. contributed equally. T.L.G. developed the study concept. J.C.P. collected the data and created the figures. R.M.B. analyzed the data and drafted the manuscript. All authors contributed to the study design and edited the manuscript.

## Competing interests

The authors declare no competing interests.
