## [Peer Review File · Nature Communications]

Reviewers' Comments:

Reviewer #1:

Remarks to the Author:

I'm pleased to say that, after two rounds of feeling confused by this paper's contribution (in part because of what I felt was a misleading take on the extant literature), the latest revisions have made things much clearer. In my previous reviews (for Nature Human Behavior), I worried that the paper described the extant literature in a false or misleading way (suggesting, for example, that categorization research doesn't use naturalistic stimuli, even though they most certainly do, eg the scene and face categorization literature). These and other presentational aspects of the paper really caused problems for me because they made the paper's contribution unclear -- it wasn't obvious what the paper was adding, given that what it claimed to adding was something that was already present. But now, after two rounds of revisions, that contribution is indeed clearer (at least to me, and hopefully to others as well). So, that main concern of mind is dealt with. Great!

As I noted in my previous review, and in a letter to the action editor when asked, I do not feel qualified to evaluate the most technical aspects of this paper. On the last round of reviews, the other reviewer raised what sounded like some very serious methodological concerns about the approach and analysis. Like I say, I don't feel that I can comment on those concerns, and whether they are addressed.

So: I am happy to recommend publication *with respect to the issues I have previously raised*. But I will hope that the journal has solicited a review from an expert on the paper's methods and analyses, since my more positive comments above cannot be taken as a wholesale endorsement of the paper (since some of it is beyond my expertise). If a more expert reviewer on those issues supports publication too, I'm very happy to join them. But if those other concerns remain, it's not my place to say whether this revision has addressed them or not.

Reviewer #2:

Remarks to the Author:

While I think the approach of combining CNN models and cognitive models is an excellent one, I have a number of concerns, questions, and comments that give me some pause in recommending publication.

- I'm not quite sure what the authors mean by "complexity" when they say that the exemplar model is more complex than the prototype model. A prototype model requires some form of abstraction at storage, which is arguable a more complex learning process than simply remembering all or a subset of experienced exemplars. An exemplar model does require more items to be stored, but as many have argued, for example Barsalou, being able to create abstractions on the fly using stored exemplars may provide more cognitive flexibility than requiring the right abstraction to be formed during learning. Now if the authors meant complexity in the sense of the work by Myung, Pitt, Navarro, and others, while it is true that exemplar models are somewhat more complex (in a

functional form sense) than prototype models, I seem to recall that the difference in that kind of complexity is not nearly as large as that between prototype and exemplar models and other kinds of models, but it has been a while since I reviewed that work. In a reply to a previous review, the authors note that exemplar models can predict "more complex ... categorization boundaries". To be clear, exemplar models do not learn complex or simple categorization boundaries, those boundaries are entirely implicit based on the nature of the similarities to exemplars from various categories. That's not a sense of "complexity" that to me would disqualify a model, except perhaps in the eye of certain beholders. There seems, perhaps, to be an attempt to appeal to some common-sense notion that "everyone" believes that exemplar models are "more complex" than prototype models, perhaps to embrace the prototype model as the "winner" in a relatively tied race of model comparison. That's not a very compelling argument statistically or theoretically, at least in my view.

- (lines 48-54) The distinctions between work in categorization and work in object recognition is not now as sharp as the authors suggest, in my opinion. I would have agreed with them 15 years ago perhaps. Not as much today. While it is true that a significant amount of past work used simple, low-dimensional, highly-controlled stimuli (in much the same way that work in attention, memory, and other domains used simple, low-dimensional, highly-controlled stimuli), a fair amount of more recent work (with objects, with faces) is using more complex real-world stimuli.

- (lines 53-56) I'm not quite sure what's gained by being so critical of the recent work by Nosofsky and colleagues (with rocks). I think that work is complementary to what's being done here (that complementarity is acknowledge in the discussion, not in the introduction). I hope that Nature Communication does not set a bar for publication that requires disparaging other recent work in order to demonstrate novelty. While the present work collects data from more subjects and more stimuli per category, the Nosofsky work has more categories of objects (30) than the present work (10), and the rocks in the Nosofsky work have a hierarchical structure.

- I'm not sure Figure 1 is necessary.

- One dimension of difference between the present work (CIFAR images) and the past work using simply, low-dimensional, highly-controlled stimuli, is that the latter often examines what is more like subordinate-level (or sub-sub-ordinate) categorization. In the classic work (and its modern instantiations) the stimuli/objects all have the same repertoire of features that differ in shape or have fairly subtle quantitative differences along particular dimensions (like the angle and spatial frequency of a gabor patch). It's like telling apart a Stratocaster from a Telecaster, or a Cabernet from a Merlot. Telling apart an airplane from a frog (in CIFAR) is like telling apart a Cabernet from milk. I guess my real point is that it is fine to have this present work stand on its own merits given the size of the human dataset and the kind of categorization task it represents and the approach the work takes combining CNN representations and cognitive models. The present work does not increase its potential impact by critiquing past work rather clumsily.

- I would also note that the nature of the difficulty of the CIFAR stimuli is different from the nature of the difficulty in the more classic experiments using simple stimuli. The CIFAR stimuli are 32x32 and are expanded to 160x160. They're highly pixelated and distorted. The difficulty is from this low pass filtering of a sort. By contrast, with an artificial set of gabors varying in orientation and spatial frequency that belong to two different categories, the difficulty doesn't stem from some filtering of the stimuli or noise masking the stimuli, but because two stimuli that belong to different categories are visually very similar in their dimensional representation (say varying in subtle way by one degree of orientation).

- pp. 11-12 The authors are likely aware of past work showing both mathematically and using

simulations the conditions under which exemplar and prototype models mimic one another (a fair amount of that by Nosofsky and by Ashby, and one by Rosseel).

- While I am confident that the authors realize that it is quite easy to come up with category structures where prototype and exemplar models perform similarly (for example, a simple family resemblance structure), much of the work addressing contrasts between prototype and exemplar models using simple stimuli has been aimed at constructing novel category structures where prototype and exemplar models make different predictions and showing that human behavior is most often more consistent with exemplar than prototype models. I am not sure that a casual reader, who is not an expert in this literature, would appreciate that important point.

- So I wonder to what extent the CIFAR images (and their categories) are really designed to distinguish prototype from exemplar models (and indeed, they seem to perform quite similarly here). Are there many instances that are near boundaries, creating the kinds of structures that McKinley and Nosofsky studies and that Ashby and Waldron studies, that led both to argue in favor of more exemplar or exemplar-like representations? In part that depends on the instances and the categories used. Images of an ostrich might easily be classified using a single prototype representation for bird when the alternative categories are cat, deer, frog, and the like. But what if there was a dinosaur? Or some other two-legged creatures? Are there El Caminos (cars) and pick-up trucks (trucks)? I just don't know if the CIFAR dataset has the kind of structure (especially given the diversity of basic-level categories, ranging from airplanes to frogs) to provide the kind of leverage needed to distinguish exemplar models and prototype models convincingly.

- Perhaps more important, to what extent are the representations learned by the CNNs (which have a logistic hyperplane on the outputs) themselves "prototype-like"? Have the complex manifolds of the objects in the CIFAR dataset been untangled over learning (in the sense of the DiCarlo TiCS paper) that it is simply enough to plop down a linear decision boundary (which is mathematically equivalent to a simple prototype model) and classify with reasonable accuracy? One of the strengths of exemplar models (that isn't highlighted in the paper) is that the same object representations (in a multidimensional psychological space) and the same exemplar representations can be used for object categorization (more abstract), object identification (more specific), category typicality, recognition memory, albeit with different weights on dimensions depending on their diagnosticity. The CIFAR object representations learned by CNN models may be specific to doing the kinds of categorizations (the 10 categories) those CNNs were trained on.

Response to Reviewers

Point-by-point response to the reviewers' comments, reproduced verbatim

Reviewer #1 (Remarks to the Author)

*I'm pleased to say that, after two rounds of feeling confused by this paper's contribution (in part because of what I felt was a misleading take on the extant literature), the latest revisions have made things much clearer [...] So: I am happy to recommend publication *with respect to the issues I have previously raised*.*

Response: We are sincerely grateful to Reviewer #1 for their patience and recommendations, and believe the manuscript to be much improved as a result.

If a more expert reviewer on those issues supports publication too, I'm very happy to join them. But if those other concerns remain, it's not my place to say whether this revision has addressed them or not.

Response: Reviewer 2, below, is such an expert, and we have endeavored to comprehensively address all of their comments, technical and otherwise, hopefully in a suitably accessible manner.

Reviewer #2 (Remarks to the Author)

While I think the approach of combining CNN models and cognitive models is an excellent one, I have a number of concerns, questions, and comments that give me some pause in recommending publication.

Response: We thank Reviewer #2 for their time and insight in reviewing the work, and hope we have addressed their comments in a satisfying manner below and in the highlighted changes to the manuscript. In general, we felt these comments fell under two themes: our coverage and portrayal of existing work, and the nature of the categorization task given the set of images for which we have collected data. To address the former, we have added a number of sections and rephrasings to the text that better communicate the literature on categorization models and offer our findings in a more complementary manner. Indeed, this was always our intent: to highlight the changes to existing approaches and debates that necessarily accompany a move to naturalistic stimuli, rather than claim any single strategy was better than others. To address the latter, we have mainly added text to the results and discussion that incorporates the comments, as well as giving our justifications below. We really do feel that using these images (and using large numbers of them) have important advantages.

Reviewer #2 (point 1): *I'm not quite sure what the authors mean by "complexity" when they say that the exemplar model is more complex than the prototype model. A prototype model requires some form of abstraction at storage, which is arguable a more complex learning process than simply remembering all or a subset of experienced exemplars. An exemplar model does require more items to be stored, but as many have argued, for example Barsalou, being able to create abstractions on the fly using stored exemplars may provide more cognitive flexibility than requiring the right abstraction to be formed during learning. Now if the authors meant complexity in the sense of the work by Myung, Pitt, Navarro, and others, while it is true that exemplar models are somewhat more complex (in a functional form sense) than prototype models, I seem to recall that the difference in that kind of complexity is not nearly as large as that between prototype and exemplar models and other kinds of models, but it has been awhile since I reviewed that work. In a reply to a previous review, the authors note that exemplar models can predict "more complex ... categorization boundaries". To be clear, exemplar models do not learn complex or simple categorization boundaries, those boundaries are entirely implicit based on the nature of the similarities to exemplars from various categories. That's not a sense of "complexity" that to me would disqualify a model, except perhaps in the eye of certain beholders. There seems, perhaps, to be an attempt to appeal to some common-sense notion that "everyone" believes that exemplar models are "more complex" than prototype models, perhaps to embrace the prototype model as the "winner" in a relatively tied race of model comparison. That's not a very compelling argument statistically or theoretically, at least in my view.*

Response: Thank you for the comment—we appreciate the opportunity to provide clarity here. Our characterization of complexity is based on statistical learning theory, and can be expressed in terms of the size of the space of functions that a decision boundary is selected from based on a sample. Prototype models historically consider only linear functions (i.e., linear boundaries), but some variants can also support quadratic functions (as mentioned in the paper). On the other hand, exemplar models are members of the class of kernel-based models, which are known to be “universal function approximators” (i.e., given enough data, they can approximate any measurable or continuous function up to any desired accuracy). The difference in complexity is therefore the difference between the space of boundaries characterized by linear and quadratic functions and the space of all possible boundaries. We have added a short sentence clarifying this assessment in the results (**lines 236-247**). Indeed, although not mentioned explicitly in, for example, Myung, Pitt, and Navarro, 2007, it is captured implicitly in the functional form term of their model complexity score, which grows with the number of exemplars.

However, it is not our intention to use this kind of complexity to argue that one class of models is better than the other. For example, linear boundaries have less representational capacity (i.e., they simply can't represent complex nonlinear boundaries); however, they require drastically less data to generalize well and suffer much less from the “curse of dimensionality”. Exemplar models have the opposite advantages and disadvantages. Further, there are relevant considerations outside of the classic framework of statistical learning (e.g., as the reviewer mentions, keeping exemplars around can be useful for on-the-fly representation construction etc.). For these reasons, we agree with the reviewer that exemplar models should not be “disqualified” on any such basis; nor would we disqualify prototype models for being fundamentally limited.

More generally, we don't see our findings as helping prototype models “win out” over exemplar models (this seems both unlikely and unhelpful), but as an interesting assessment of alternate strategies in a novel context. Finally, we agree that exemplar models represent decision boundaries only implicitly, and this is also true for prototype models. However, implicit or explicit, the correspondence between a density estimator and its class of corresponding boundaries is exact and well-understood (e.g., each corresponds to a Bayes optimal decision boundary, as established by Ashby & Alfonso-Reese, 1995).

While space is more limited than we would like, we have revised and added references to the second paragraph of the introduction (**lines 36-42**), and elsewhere draw attention to the advantages of exemplar models that are not encompassed by the framework discussed above.

Reviewer #2 (point 2): *(lines 48-54) The distinctions between work in categorization and work in object recognition is not now as sharp as the authors suggest, in my opinion. I would have agreed with them 15 years ago perhaps. Not as much today. While it is true that a significant amount of past work used simple, low-dimensional, highly-controlled stimuli (in much the same way that work in attention, memory, and other domains used simple, low-dimensional, highly-controlled stimuli), a fair amount of more recent work (with objects, with faces) is using more complex real-world stimuli.*

Response: To clarify, while there is increasing empirical work using more naturalistic stimuli in both of these literatures, the theoretical literature on cognitive models of categorization has almost exclusively focused on experiments with simplistic stimuli. This makes sense – these models require identifying the features of stimuli, and cannot simply be applied in pixel space directly. Our goal is to bring theoretical work on cognitive models of categorization into line with the advances that have been made in other areas of cognitive science and neuroscience using naturalistic stimuli. We discuss this point in the third paragraph of the introduction (**lines 52-65**).

Reviewer #2 (point 3): *(lines 53-56) I'm not quite sure what's gained by being so critical of the recent work by Nosofsky and colleagues (with rocks). I think that work is complementary to what's being done here (that complementarity is acknowledge in the discussion, not in the introduction). I hope that Nature Communication does not set a bar for publication that requires disparaging other recent work in order to demonstrate novelty. While the present work collects data from more subjects and more stimuli per category, the Nosofsky work has more categories of objects (30) than the present work (10), and the rocks in the Nosofsky work have a hierarchical structure.*

Response: We regret that our review of the work may have sounded too critical, especially since we follow and greatly appreciate such work. To address these concerns, we have revised the final part of the third paragraph in the introduction (**lines 60-66**), revised the second paragraph of the section “A naturalistic image dataset” to highlight their dataset and remove the unhelpful comparison (**lines 141-143**), and revised our comparison of modeling approaches in the discussion to offer a more forward-looking synthesis (**lines 411-432**).

Reviewer #2 (point 4): *I'm not sure Figure 1 is necessary.*

Response: If this were a traditional psychology journal, we would see Reviewer #2's point. However, because of the broad intended audience of Nature Communications, we expect that informing the readers on existing research methods and results and the nature of our contribution in multiple ways will be important; Figures 1 helps with

this. Having said this, and pending recommendations from the Editor, we are happy to work with Reviewer #2 to adapt it to something they consider more suitable for this purpose or remove it altogether.

Reviewer #2 (point 5): *One dimension of difference between the present work (CIFAR images) and the past work using simply, low-dimensional, highly-controlled stimuli, is that the latter often examines what is more like subordinate-level (or sub-sub-ordinate) categorization. In the classic work (and its modern instantiations) the stimuli/objects all have the same repertoire of features that differ in shape or have fairly subtle quantitative differences along particular dimensions (like the angle and spatial frequency of a gabor patch). It's like telling apart a Stratocaster from a Telecaster, or a Cabernet from a Merlot. Telling apart an airplane from a frog (in CIFAR) is like telling apart a Cabernet from milk. I guess my real point is that it is fine to have this present work stand on its own merits given the size of the human dataset and the kind of categorization task it represents and the approach the work takes combining CNN representations and cognitive models. The present work does not increase its potential impact by critiquing past work rather clumsily.*

Response: We appreciate the insight and agree that our characterization of past work was in need of revision. We have updated the manuscript accordingly: we more clearly highlight our categories as basic-level when discussing our stimuli in the section entitled "A naturalistic image dataset" (**lines 143-149**).

Having said this, over all the images in CIFAR it is not quite clear that there aren't also many Cabernet-Merlot type comparisons. For example, on inspecting birds and planes, many of them do share the same feature set, with subtle variations of degree (the pairs of categories for which this is true can be seen in the confusion matrix). That is to say, perhaps using larger naturalistic datasets common to other fields will always mean averaging across some discrepancy in the meaning or level of category labels across pairs of images. This is a good direction to explore in future work based on the techniques we and others are proposing, and so in the discussion we have added a paragraph suggesting this; for example, by replicating the study with a subset of the lower-level categories of ImageNet (**lines 437-439**).

Reviewer #2 (point 6): *I would also note that the nature of the difficulty of the CIFAR stimuli is different from the nature of the difficulty in the more classic experiments using simple stimuli. The CIFAR stimuli are 32x32 and are expanded to 160x160. They're highly pixelated and distorted. The difficulty is from this low pass filtering of a sort. By contrast, with an artificial set of gabors varying in orientation and spatial frequency that belong to two different categories, the difficulty doesn't stem from some filtering of the stimuli or noise masking the*

stimuli, but because two stimuli that belong to different categories are visually very similar in their dimensional representation (say varying in subtle way by one degree of orientation).

Response: This is a fair point, and we have added text explaining this distinction in the stimulus description section entitled "A naturalistic image dataset" (**lines 168-170**). On the other hand, as noted in the response above there are certainly many pairs of images for which the discrimination is similar to this; in the end we are averaging over both forms of uncertainty.

Reviewer #2 (point 7): *pp. 11-12 The authors are likely aware of past work showing both mathematically and using simulations the conditions under which exemplar and prototype models mimic one another (a fair amount of that by Nosofsky and by Ashby, and one by Rosseel).*

Response: Thank you for bringing these to our attention. We have added a paragraph to the Discussion (**lines 401-403**) indicating the relevance of these contributions for future work.

Reviewer #2 (point 8): *While I am confident that the authors realize that it is quite easy to come up with category structures where prototype and exemplar models perform similarly (for example, a simple family resemblance structure), much of the work addressing contrasts between prototype and exemplar models using simple stimuli has been aimed at constructing novel category structures where prototype and exemplar models make different predictions and showing that human behavior is most often more consistent with exemplar than prototype models. I am not sure that a casual reader, who is not an expert in this literature, would appreciate that important point.*

Response: Thank you for bringing this to our attention. We have revised the second paragraph of the introduction (**lines 33-36**), which before had only described the effort to design discriminatory stimuli, but not the purposeful use of novel items.

Reviewer #2 (point 9): *So I wonder to what extent the CIFAR images (and their categories) are really designed to distinguish prototype from exemplar models (and indeed, they seem to perform quite similarly here). Are there many instances that are near boundaries, creating the kinds of structures that McKinley and Nosofsky studies and that Ashby and Waldron studies, that led both to argue in favor of more exemplar or exemplar-like representations? In part that depends on the instances and the categories used. Images of an ostrich might*

easily be classified using a single prototype representation for bird when the alternative categories are cat, deer, frog, and the like. But what if there was a dinosaur? Or some other two-legged creatures? Are there El Caminos (cars) and pick-up trucks (trucks)? I just don't know if the CIFAR dataset has the kind of structure (especially given the diversity of basic-level categories, ranging from airplanes to frogs) to provide the kind of leverage needed to distinguish exemplar models and prototype models convincingly.

Response: We agree that the question of whether CIFAR-10 (relative to other possible natural image sets) is discriminative of candidate models is very important, but answering such a question is nontrivial. For example, there are clearly many images which, on visual inspection, appear more or less equally probable under two classes. As highlighted above, this can be seen in the confusion matrix in Figure 3. As a first approximation to the number of borderline cases, the tail of the human-guess entropy histogram presented in Figure 3 could be used, for a comparison between at least 2 of the categories. This would contain around 1000 images (10%)—presumably enough to offer some discriminability. Along with the current manuscript, we hope to host the data with the journal's dataset arm, and plan to include many more in-depth descriptive statistics of this nature that will allow follow up work to rigorously address this question.

More broadly, a number of difficult questions arise in the context of assessing the number of borderline images. For example, how do we define a threshold for what is discriminative? One of the benefits of very large datasets (that we hope to espouse) is that they allow us to effectively average over what can often be noisy comparisons between models (which may appear highly discriminative) on smaller subsets of data. However, this does not by any means rule out more discriminative yet still large datasets, which we hope to see emerge in the future, nor does it discount past work without large datasets. Our hope is that the additional dimensions of featurization, dimensionality, and stimulus sample size can be better understood and integrated into future studies of human categorization. We also note that CIFAR-10 categories have been mapped to nodes of the WordNet hierarchy, and already come packaged with subordinate-level labels. This will allow future work to probe lower-level category learning behavior, and compare directly to the upstream categories that our to-be-public data will provide.

Reviewer #2 (point 10): *Perhaps more important, to what extent are the representations learned by the CNNs (which have a logistic hyperplane on the outputs) themselves "prototype-like"? Have the complex manifolds of the objects in the CIFAR dataset been untangled over learning (in the sense of the DiCarlo TiCS paper) that it is simply enough to plop down a linear decision boundary (which is mathematically equivalent to a simple prototype model) and classify with reasonable accuracy? One of the strengths of exemplar models (that isn't highlighted in the paper) is that the same object representations (in a multidimensional psychological space) and the same exemplar representations can be used*

for object categorization (more abstract), object identification (more specific), category typicality, recognition memory, albeit with different weights on dimensions depending on their diagnosticity. The CIFAR object representations learned by CNN models may be specific to doing the kinds of categorizations (the 10 categories) those CNNs were trained on.

Response: These questions are interesting and constitute worthwhile follow-ups. We have added a note tempering strong conclusions based on the architecture / learned representations of the CNN in the results (**lines 352-359**); although, over the range of representations we assess this bias is not generally true, whereas the equivalence in model performance still holds (discussed in **lines 259-279** and **357-359**). We have also discussed the important advantages of exemplar models in the second paragraph of the introduction (**lines 38-42**), which provides strong motivation for future model comparison with a more diverse suite of benchmarks (e.g., typicality). Indeed, we are currently in the process of collecting a corresponding typicality dataset for the same 10,000-image set that researchers can use in the future to (re-)probe some of these questions going forward, and we expect exemplar models to continue to exhibit unique advantages. Another follow-up project that is currently underway proposes and evaluates CNNs designed explicitly to support the various functions and advantages of classic exemplar models (since, as the reviewer mentions, CNN classifiers are currently biased to linearly separate classes—although they need not correspond to the category feature means). Projects like these, along with those of the greater research community with which our datasets will be shared, will hopefully help to make meaningful progress on these lingering questions in the coming years.

Reviewers' Comments:

Reviewer #2:

Remarks to the Author:

The authors addressed my comments.